# Omeprazole use and risk of chronic kidney disease evolution

**João Victor Marques Guedes**[1☯]*, **Jéssica Azevedo Aquino**[1☯], **Tássia Lima Bernardino Castro**[1☯], **Flávio Augusto de Morais**[2☯], **André Oliveira Baldoni**[1☯], **Vinícius Silva Belo**[1☯], **Alba Otoni**[1☯]

**1** Department of Health Sciences, Federal University of São João Del-Rei (UFSJ), Divinópolis, Minas Gerais, Brazil, **2** Department of Nephrology, Ambulatory of Municipal Polyclinic of Divinopolis, Divinópolis, Minas Gerais, Brazil

☯ These authors contributed equally to this work.

\* jvmg92@gmail.com

**Data Availability Statement:** All relevant data are within the manuscript and its Supporting Information files.

**Funding:** The author(s) received no specific funding for this work.

## Abstract

### Rationale, aims and objectives

In recent years, the use of proton pump inhibitors (PPI), especially omeprazole, has been associated with development of chronic kidney disease (CKD). These drugs are widely used worldwide. Although some studies have found an association between the use of PPI and the onset of acute renal failure and CKD. This study aims to analyze the association between the continuous use of omeprazole and the progression of CKD in adult and elderly individuals.

### Method

A retrospective cohort study was conducted with patients followed up at a nephrology clinic in Brazil, in 2016 and 2017. Information about clinical and sociodemographic data, health behaviors, and medication use were collected from all patients diagnosed with CKD through consultation of medical charts and the Brazilian health information system (SIS). The participants were allocated into two groups: users and non-users of omeprazole, and the progression of CKD was then evaluated for each group. In the bivariate analysis, the Mann-Whitney U test to compare the quantitative variables between groups, and the Pearson/Fisher two-tailed chi-square test to compare the categorical variables were applied. Multivariate analysis was performed using Cox regression.

### Results

A total of 199 CKD patients were attended in the polyclinic, and of these, 42.7% were omeprazole users. There was a higher percentage of CKD progression in users (70.6%) compared to non-users (10.5%). The hazard ratio was 7.34 (CI: 3.94–13.71), indicating a higher risk of progression to worse stages of CKD in omeprazole users than in non-users. As for the other variables, no statistically significant difference was found between groups (p > 0.05).

**Competing interests:** The authors have declared that no competing interests exist.

## Conclusion

An association between omeprazole use and progression of CKD stage was identified, showing a higher risk of disease evolution among omeprazole users.

## Introduction

Chronic kidney disease (CKD), characterized by progressive deterioration of biochemical and physiological functions of the body systems, can be defined as a syndrome caused by the progressive decrease in renal function [1–4].

In Brazil about 10 million people present some level of renal dysfunction, and the worldwide incidence of renal disease increases around 10% per year [5]. Because of the changes in life expectancy that have made the elderly become the dominant group in populations worldwide, the epidemiological figures of CKD are increasing continually, placing renal diseases in the epidemic category. This is a troubling situation and needs to be quickly controlled to try to minimize the devastating impacts of CKD, not only for patients, but also for public managers who have to invest large amounts of money in the attempt to provide quality treatment for those affected [6].

Recently, parallel to the classical causes of development and progression of CKD, Diabetes Mellitus (DM) and Systemic Arterial Hypertension (SAH), the use of drugs which were once considered safe have been identified as possible causes of renal damage. Among these drugs, proton pump inhibitors (PPI), highly prescribed worldwide to treat gastroesophageal reflux and peptic ulcers through inhibition of gastric acid synthesis, have shown to be closely associated with acute interstitial nephritis (AIN), reduction of glomerular filtration rate (GFR) and the development of CKD [7–11]

Initially it was believed that these drugs were associated with the development of recurrent acute renal injury, generating an AIN process that could be chronic and cause loss of renal function by successive renal tissue regeneration [7,12,13]. However, recent studies found that loss of renal function is not necessarily caused by sequential acute lesions, as the use of PPI has been associated with CKD regardless of the occurrence of previous acute episodes [8,10].

The association between PPI use and CKD development is a recent and not fully understood topic. Only a few studies evaluating the impact generated by the use of these drugs, specifically on CKD progression and staging, have been performed. In addition, considering the high consumption of PPI in Brazil [14] and in the world population, as well their adverse effects, it is necessary to conduct researches to better elucidate this association. Such research may favor the creation of clinical protocols to promote the rational use of PPI, without disregarding the cost and benefit of this therapeutic approach. In this sense, the present study aims to analyze the association between the regular use of PPI and CKD progression in adult and elderly individuals, as well to analyze the survival of these patients.

## Methods

### Ethical aspects

This study was approved by the Research Ethics Committee of the Federal University of São João del-Rei (UFSJ) Dona Lindu Center-West Campus CCO (CPEC: 65858117.3.0000.5545/n° 2.010.528).

## Study design and population

This is a retrospective cohort study developed at the nephrology ambulatory sector of the Municipal Polyclinic of Divinópolis-MG, Brazil, from March 2017 to March 2019, following the STROBE checklist for observational research. Eligibility criteria were defined as all adult and elderly patients of both genders with established CKD diagnosis who were already undergoing clinical follow-up or who were diagnosed after insertion in the nephrology ambulatory clinic in the years 2016 and 2017. During this period, 241 patients were attended. Of these, 42 patients were excluded due to lack of confirmation of CKD diagnosis (28 cases), absent creatinine values (9 cases) and history of neoplasms (5 cases). Patients without baseline creatinine values (T0) were excluded from the study because it would not be possible to determine the estimated glomerular filtration rate (eGFR) and thus the baseline CKD stage. Similarly, those patients who had not recorded serum creatinine values at subsequent times (T1, T2, T3 and T4) were also excluded. Patients who underwent previous kidney transplantation and/or patients on renal replacement therapy, and patients who had missing data related to some of the variables used during the two-year follow-up were not included in this study. Thus, a total of 199 patients with CKD eligible for the study were included.

For all eligible patients, follow-up information was recorded for a period of two years after the date of insertion in the nephrology service. Baseline data was collected for population characterization at the initial moment (T0), and information about CKD staging of each patient included in the study was collected at five further times, at intervals of six months (T1, T2, T3 and T4).

## Data collection

All variables were collected from two secondary sources: medical charts and the Brazilian health information system (SIS). The medical charts were evaluated for the presence of CKD diagnosis, for inclusion of patients in the study. After confirmation of the diagnosis, other variables were collected from this source: age in full years; gender; systolic and diastolic blood pressure (categorized as normal ≤ 120 / ≤ 80 mmHg and altered > 120 / > 80 mmHg) [15]; serum creatinine; eGFR; classification of CKD stage [4]; presence and/or absence of comorbidities; and prescribed medicines.

After collection of data from medical charts, data were collected from the SIS at baseline for complementation and confirmation of sociodemographic, occupational and clinical information and health-related behaviors: schooling; marital status; race; occupation; diagnosis of dyslipidemia; smoking; alcohol consumption; sedentary lifestyle; and dispensed medicines.

## Exposure variable: Omeprazole use

Among the PPI drugs available, only omeprazole was used by the patients evaluated in this study (daily dose of 20 mg). This can be justified because it is the only PPI available in the public pharmacies of Divinópolis-MG.

Regular users of omeprazole were considered those patients who had this medicine registered in the medical records for a period of three months or more during the data collection period. Such classification of the participants as current PPI users when the time of use was three months or more was based on the consultation of previous studies [8,16]. Initially, users of omeprazole were considered as those who at baseline (T0) had used this drug previously for at least three months. Thereafter all patients defined as users were evaluated for the other times (T1, T2, T3, T4) for the use of omeprazole, and only those who maintained its use during all evaluation times for at least two years were included. Follow-up data on drug use were available for all follow-up times. Only patients who continued to use the drug during all assessment times were considered exposed to omeprazole. In the two years of follow-up, there was no

change in the prescription for any of the patients included, maintaining the daily dose of 20 mg/ day. Those who discontinued omeprazole during follow-up were not included in the study.

## Outcome variable: CKD stage evolution

In this study only patients with confirmed diagnosis of CKD were included, patients with acute kidney injury (AKI) and other renal diseases were not evaluated.

The evaluation of CKD stage and the identification of omeprazole users and non-users occurred at the time of diagnosis in the ambulatory clinic, followed by subsequent consultations. For a large number of patients, the CKD stage was available and recorded by the prescriber in the medical record at the time of the consultation. When the CKD stage was not available, the eGFR was calculated and classified according to the KDIGO [4] and MS [17] guidelines using the CKD-EPI equation without correction for race [18,19]. Evolution of CKD was considered when the stage changed to a more advanced stage of renal impairment between one consultation to another, considering an increasing scale of severity starting from stage 1. In addition to this analysis, for each time, a 25% or more reduction in eGFR compared to the eGFR of the previous time was evaluated. Plus, a rapid progression analysis (5 mL/1.73m$^2$/ year) was performed as recommended by KDIGO [4].

## Explanatory variables: Use of nephrotoxic and nephroprotective drugs

In addition to PPI, the use of other drugs classically defined as nephrotoxic and nephroprotective agents was analyzed. Angiotensin-converting enzyme (ACE) inhibitors and angiotensin II receptor blockers (ARBs), included in current pharmacological nephroprotective therapy for patients with diabetic nephropathy and CKD, are recommended as nephroprotective agents according to the KDIGO [4] and MS [17] guidelines for the treatment of CKD patients. As for nephrotoxic drugs, non-steroidal anti-inflammatory drugs (NSAIDs) have been classically described as drugs with confirmed nephrotoxicity [20].

Despite the proposed collection regarding the use of other nephrotoxic drugs (aminoglycosides, lithium, amphotericin, acyclovir, foscarnet, polymyxins, glycopeptides, tacrolimus, cyclosporine, clopidogrel and interferon), only NSAIDs were prescribed/dispensed to the study population during the collection period. Thus, patients using these drugs were also evaluated, even patients using low-dose aspirin to prevent cardiovascular events.

The use of non-prescribed or drugs not reported at medical consultations was also evaluated using medicine dispensing reports in public pharmacies, generated by the Brazilian health information system (SIS), during the follow-up period.

## Database design and processing

After all data were collected, a database was constructed and filled using Epidata software version 3.1. Data were double typed. The Kappa agreement test was applied to validate the database, resulting in a Kappa coefficient of 0.92, reflecting an almost perfect agreement (0.81–1.00) [21]. Data were exported to the Statistical Package for the Social Sciences (SPSS) version 20 for statistical analysis. The STATA SE version 12.0 software was used for Cox regression analysis.

## Statistical analysis

The study population was characterized through descriptive analysis. The Shapiro-Wilk test was applied to check the normality of the quantitative variables. The median and 25th and

75th percentiles were calculated, since all variables presented non-normal distribution. Relative frequency data were presented for the categorical variables. The Mann-Whitney U test and the Pearson/Fisher two-tailed chi-square test were used in the bivariate analysis.

Because this is a cohort study with a two-year follow-up and the defined outcome was the evolution of CKD to worse stages, Cox regression was used for the multivariate model to investigate the influence of time on the strength of association between omeprazole use and CKD evolution [22]. In order to determine the variables to be used in the multivariate analysis, univariate analysis using Cox regression was initially made, and variables that presented p-value < 0.20 were considered eligible for the multivariate model. Thus, the following variables were used in the fitting of the multivariate analysis: omeprazole use and ARBs use.

After the multivariate analysis, Kaplan-Meier estimates were used to evaluate the evolution of risk to omeprazole users and non-users as a function of time.

## Results

The population of CKD diagnosed patients followed up at the nephrology ambulatory clinic of the Municipal Polyclinic of Divinópolis-MG, Brazil, during the years 2016 and 2017 was 204 people (37 excluded cases without CKD diagnosis). During this period, 406 patients attended consultations, resulting in a CKD prevalence of 50.24% in the nephrology ambulatory. The total of patients eligible for the study was 199, classified as omeprazole users (85) and non-users (114).

Of the patients evaluated, 64.5% lived with their partners, 51.7% were of the black/brown race, and 37.2% were retired. The most prevalent comorbidities were SAH (89.4%) and DM type II (39.7%).

The main sociodemographic and clinical characteristics and health behaviors of patients are described in Table 1. A higher frequency of patients with altered blood pressure was found in the omeprazole group, and this difference was statistically significant (p-value = 0.016). No statistically significant difference was found between groups for the other variables, showing similarity between the groups studied.

The evaluated patients presented a median of 6.0 (4.0–8.0) for total medication use. Regarding the use of nephrotoxic and nephroprotective drugs, a frequency of 3.5% was found for NSAIDs, 21.6% for ACE inhibitors, and 61.3% for ARBs. No significant statistical difference was found between groups for these variables.

There was a significant difference in CKD evolution between omeprazole users and non-users (p < 0.0001) (Table 2). Confirming a higher CKD evolution in omeprazole users, a decrease in eGFR ≥ 25% was found in omeprazole users more frequently than in non-users, and this difference was statistically significant (p = 0.002). Regarding the rapid progression of CKD (5 mL/1.73m2/year), no statistically significant differences were found between users and non-users of omeprazole (p = 0.233).

The covariable ARBs use presented a p-value < 0.20 and was then used to adjust the multivariate analysis through Cox regression. The use of the other drugs evaluated (NSAIDs, ACE inhibitors, ARBs) and blood pressure presented p-value> 0.20 in the univariate Cox analysis, showing that these are not confounding variables and, therefore, not included as adjustment variables in the multivariate analysis. Initially, an unadjusted hazard ratio (HR) of 6.72 was obtained. After adjustment, a HR of 7.34 (CI 3.94–13.71) was found, indicating a risk of CKD progression approximately 7.4-fold higher in omeprazole users when compared to non-users. All adjustments made were statistically significant (p-value < 0.05) (Table 3).

Fig 1 shows the Kaplan-Meier survival plot for the worsening of CKD stage in omeprazole users and non-users. There was a higher risk of progression to worse stages among omeprazole

**Table 1. Sociodemographic and clinical data and health behaviors of omeprazole users and non-users followed up at the nephrology ambulatory clinic of the Municipal Polyclinic of Divinópolis-MG, Brazil, 2016–2017 (n = 199).**

| Variables | General (n = 199) | Non-omeprazole users (n = 114) | Omeprazole users (n = 85) | p-value |
|---|---|---|---|---|
| **Gender** | | | | 0.409* |
| Male | 54.3% | 51.8% | 57.6% | |
| **Age** | 72 (62.0–80.0) | 74 (62.5–80.0) [#] | 70 (61.0–79.5) [#] | 0.297** |
| **Schooling** | | | | 0.884* |
| Illiterate | 12.3% | 12.9% | 11.3% | |
| Incomplete elementary school | 64.5% | 63.4% | 66.1% | |
| Complete elementary school | 11.6% | 12.9% | 9.7% | |
| Others | 11.6% | 10.8% | 12.9% | |
| **Blood pressure (S/D mmHg)** | | | | 0.016* |
| Normal (≤120/≤80) | 72.4% | 78.9% | 63.5% | |
| Altered (>120/>80) | 27.6% | 21.1% | 36.5% | |
| **Serum creatinine** | 1.60 (1.40–2.00) | 1.60 (1.40–2.05) [#] | 1.59 (1.40–1.99) [#] | 0.766** |
| **eGFR** | 37.0 (26.4–47.0) | 35.8 (26.0–45.0) [#] | 38.9 (28.0–47.7) [#] | 0.215** |
| **Dyslipidemia** | | | | 0.329* |
| Absent | 81.9% | 84.2% | 78.8% | |
| Present | 18.1% | 15.8% | 21.2% | |
| **Alcohol consumption** | | | | 0.579* |
| No | 82.6% | 80.0% | 85.2% | |
| Yes | 6.0% | 8.3% | 3.7% | |
| Ex-consumer | 11.4% | 11.7% | 11.1% | |
| **Smoking** | | | | 0.883* |
| No | 76.3% | 77.6% | 74.5% | |
| Yes | 17.5% | 17.2% | 18.2% | |
| Ex-Smoker | 6.2% | 5.2% | 7.3% | |
| **Sedentary lifestyle** | | | | 0.225* |
| No | 66.8% | 71.9% | 60.0% | |
| Yes | 33.2% | 28.1% | 40.0% | |
| **CKD stage** | | | | 0.328* |
| Stage 2 | 11 (5.5%) | 7 (6.1%) | 4 (4.7%) | |
| Stage 3a | 62 (31.1%) | 31 (27.2%) | 31 (36.5%) | |
| Stage 3b | 71 (35.7%) | 42 (36.8%) | 29 (34.1%) | |
| Stage 4 | 53 (26.6%) | 32 (28.1%) | 21 (24.7%) | |
| Stage 5 ND | 2 (1.1%) | 2 (1.8%) | 0 (0.0%) | |

[#]Median (25th Percentile– 75th Percentile).

*Two-tailed Pearson chi-square test.

**Non-parametric Mann-Whitney U test. S: Systolic; D: Diastolic; eGFR: Estimated glomerular filtration rate; ND: Non-dialytic.

users. At six months of follow-up, the risk of progression among omeprazole users was approximately 36.0%, while among non-users this risk was approximately 5.0%. It was observed that after the two-year follow-up, the risk of progression to a worse stage of CKD was approximately 84.0% among omeprazole users, while it was approximately 18.0% among non-users.

## Discussion

A statistically significant association was found between regular use of omeprazole and CKD progression in patients assisted at the nephrology ambulatory clinic (p-value < 0.001), with a

**Table 2. Relative frequency of CKD evolution per stage of users and non-users of omeprazole followed-up at the nephrology ambulatory clinic of the Municipal Polyclinic of Divinópolis-MG, Brazil, 2016–2017 (n = 199).**

| CKD evolution | General (n = 199) | Non-omeprazole users (n = 114) | Omeprazole users (n = 85) | p-value |
|---|---|---|---|---|
| Total | 36.2% | 10.5% | 70.6% | < 0.0001* |
| Stage 2 → Stage 3a | 2.4% | 0.0% | 5.5% | |
| Stage 3a → Stage 3b | 16.9% | 7.6% | 29.5% | |
| Stage 3b → Stage 4 | 12.9% | 2.9% | 26.4% | |
| Stage 4 → Stage 5 ND | 3.9% | 0.0% | 9.2% | |
| Stage 5 ND → Stage 5 D | 0.0% | 0.0% | 0.0% | |

* Two-tailed Pearson chi-square test; CKD: Chronic kidney disease; ND: Non-dialytic.; D: Dialytic.

higher frequency of CKD progression for omeprazole users compared to non-users. After adjustments made in the multivariate analysis, a HR of 7.34 was obtained, indicating an approximately seven-fold higher risk of progression to a worse stage among patients using PPI. There is a limitation in the current literature with respect to studies associating CKD evolution and PPI use. A considerable number of studies only reported that long-term exposure to these drugs favors the onset of acute and also CKD, but they did not evaluate the progression of CKD in patients [7,8,10,23,24].

Investigating the possible mechanisms associated with renal damage caused by the use of PPI, Kamal et al. (2018) identified that the use of these drugs may be associated with the installation of a chronic process arising from a recurrent acute process, as well as the installation of the chronic process independent of the installation of AKI. In this sense, the authors propose some mechanisms that may be involved: the deposition of the drug and its metabolites in renal tissue, which may culminate in renal interstitial fibrosis, leading to chronicity of the lesion and onset of CKD; reduced nitric oxide synthesis, caused by inhibition of the proton pump of cell lysosomes, and thus production of highly reactive superoxide anion, which causes renal endothelial dysfunction; and hypomagnesemia, as low magnesium levels are able to increase secretion of atherogenic and inflammatory substances, producing endothelial dysfunction of the renal tissue [25].

In contrast to that presented for PPI, no results have yet been found that associate the use of H2 antagonists with the installation and/or progression of CKD. Some previous studies have suggested a possible association between the use of these drugs and the setting up of acute processes [26,27], but these results are still controversial, as current studies have not found an association between the use of these drugs and the installation of acute or chronic processes [8, 9, 28]. Possibly this difference found between the use of PPI and H2 antagonists may be related

**Table 3. Multivariate analysis using Cox regression for association between evolution of chronic kidney disease and omeprazole and ARBs use in patients followed-up at the nephrology ambulatory clinic of the Municipal Polyclinic of Divinópolis-MG, Brazil, 2016–2017 (n = 199).**

| Variables | Hazard Ratio (HR) | Confidence Interval (95% CI) | p-value |
|---|---|---|---|
| **Omeprazole use** | | | |
| Non-users (reference category) | 1.00 | - | - |
| Users | 7.34 | 3.94–13.71 | < 0.001* |
| **ARBs use** | | | |
| Non-users (reference category) | 1.00 | - | - |
| Users | 0.60 | 0.37–0.96 | 0.033* |

* p-value < 0.05

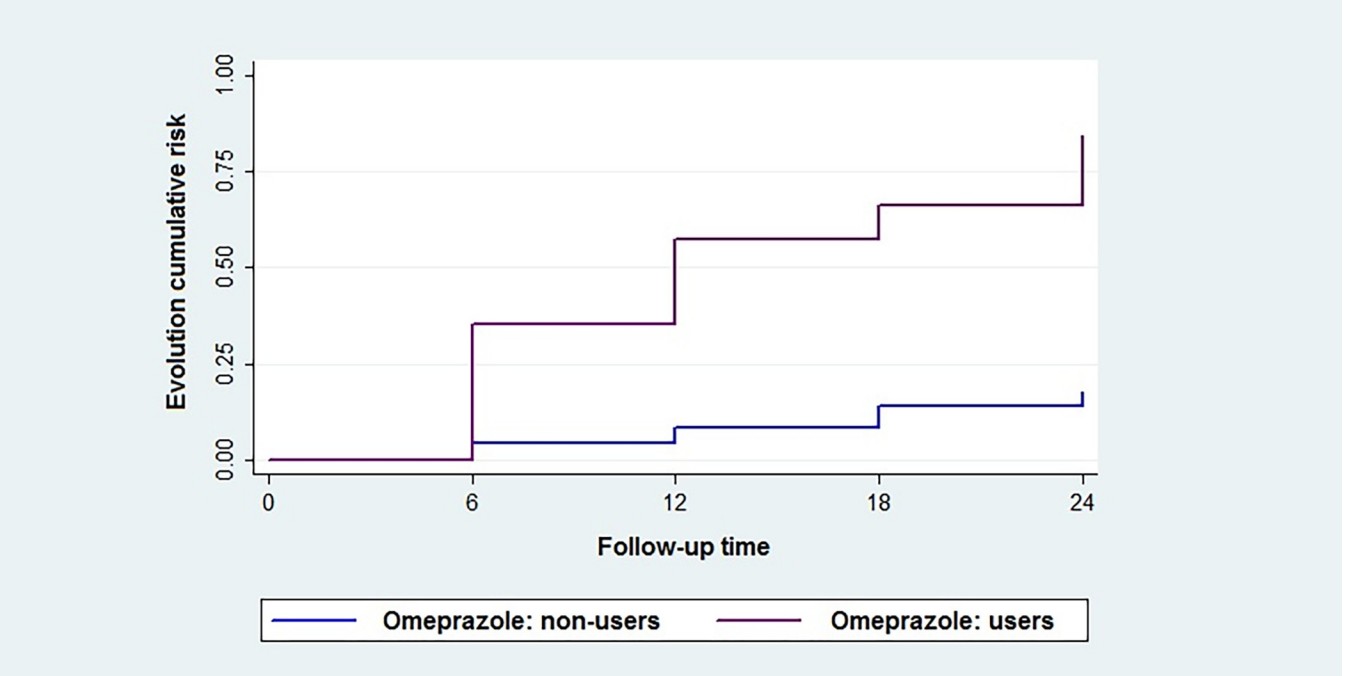

**Fig 1. Kaplan-Meier estimates for CKD progression among omeprazole users and non-users.** Fig 1 shows the Kaplan-Meier survival plot for the occurrence of CKD evolution to worse stages in omeprazole users and non-users. The follow-up time is shown in the X axis and the cumulative risk of CKD evolution in the Y axis.

to their respective mechanisms of action, since they are of different drug classes and trigger different physiological effects.

Regarding the deposition of drugs and their metabolites in renal tissue, H2 antagonists have already been mentioned, and possibly by this mechanism could be associated with acute processes, however, it should be noted that at initially a large number of drugs could trigger an inflammatory response in renal tissue [27]. Regarding nitric oxide synthesis, there are no studies to date that found an association between the reduction of its synthesis and H2 antagonists, an expected result since these drugs do not act by inhibiting proton pumps, but by blocking histamine receptors. (H2). Regarding hypomagnesemia, a population study conducted by Markovits et al (2014) found that the use of H2 antagonists is associated with mild hypomagnesemia, only in use for long periods, and found no association of these drugs with moderate or severe hypomagnesemia. For PPI, a significant association was found with a higher association strength for mild, moderate and severe hypomagnesemia [29]. Thus, it can be concluded that probably the mechanisms leading to the onset and progression of CKD by the use of PPI are associated with the onset of hypomagnesemia, as already proposed by Kamal et al. (2018) [25]. In fact, a variety of studies have found a significant association between PPI use and hypomagnesemia [28–32].

Hypomagnesemia associated with the use of PPI may be partly explained by the renal loss of magnesium due to decreased resorption as well as decreased absorption, as PPI are capable of altering the pH of the gastrointestinal tract, thereby decreasing absorption mediated by TRPM6 and TRPM7 transporters [32].

These findings, supported by what has already been described in the literature, draw attention to the fact that PPI nephrotoxicity is a risk not only for the initial onset of renal tissue damage but for a progressive loss of renal function when CKD is already present before the use

of these drugs. In this sense, the present study brings results not yet explained in the current literature. Considering the present findings, it is necessary to rethink the use of these drugs in any CKD stage, as this can be an independent factor for worsening of renal function. Most worrying is that currently there is no restriction in this aspect in the guidelines for treatment of CKD patients, for either MS [17] or KDIGO [4]. It is important that the guidelines for treatment of CKD patients be updated, alerting them about these new risk factors and about the need for rational and cautious use of PPI.

In addition to the guidelines, immediate awareness about the harm caused by regular and continuous use of PPI is needed not only on the part of health professionals but also of CKD patients. This is necessary in particular because of the easy acquisition, at low cost, and the high frequency of inappropriate use of this drug, as already pointed out in the literature, with reports of use without clinical indication in 40–80% of cases [33,34].

Regarding the sociodemographic characteristics, the patients in the present study were mostly male (54.3%) and of the white race (46.3%). According to Cobo et al., as happens in other diseases, the gender of the patient is of fundamental importance for the approach to CKD patients, because men and women differ in renal physiology and in the basic pathophysiology, complications, signs and symptoms of this disease, thus requiring a differentiated approach according to gender [35]. In general, females present greater renovascular resistance, lower absolute GFR, and lower renal flow than males [36], not to mention differences in the functional physiology of the renin angiotensin system and blood pressure control [37,38]. Thus, it is necessary to emphasize the approach to female patients, in spite of all the risks for cardiovascular diseases (CVD).

Gender should be taken into account as a potential factor for increased risk of CVD. A precautionary and comprehensive approach is necessary without losing sight of the whole context involving CKD evolution and the risk of death from CVD. In none of these studies were found associations of gender with PPI use, or with the CKD evolution. In contrast, when it comes to the physiological loss of renal function with the advance of age, the female gender has shown to offer greater protection, which is explained by gender hormones and nitric oxide regulation, and consequent less reduction of renal aging [39]. As for race, in the United States of America, it is observed that African Americans present higher rates of renal diseases when compared to the other races [40]. However, in Brazil, relating race to renal function may not be prudent because there is a great racial miscegenation inherent in the national historical-cultural factor. In the present study, a number of studies conducted on Brazilian populations showed that race adjustment is not necessary for eGFR calculation, for example, because this variable does not interfere with the final results [41–43].

Regarding age group, the population assisted at the nephrology ambulatory clinic was mainly composed of elderly people (81.8%). This is worrisome because it implies two simultaneous risk factors: advanced age and PPI use without any level of restriction [44]. Perhaps this approach adopted by the ambulatory clinic so far has been based on the lack of regulations to guide a restrictive approach to PPI use.

In Beers criteria, which include actions related to the rational and safe use of drugs in elderly patients, there is no previous warning about the risk of renal damage due to PPI use in the elderly population [45]. This situation needs to be discussed because these drugs are frequently used in this age group and for long periods [46]. Furthermore, polypharmacy is usually common among elderly patients, which further corroborates the use of PPI to ensure protection of the gastric mucosa [47]. However, in the present study, no statistically significant difference was found in the age variable between groups of users and non-users of PPI (p > 0.05).

Most patients assisted at the nephrology ambulatory clinic had incomplete primary education (64.5%), lived with their partners (64.7%), and were retired (37.2%). The low schooling

found in this study for a population that is composed mainly of the elderly individuals is worrisome, because this population needs more health care and makes frequent use of a great quantity of medicines. Of course, the simple identification of the low level of education of these patients does not indicate that they are incapable of accessing health services, but it represents an increased likelihood of difficulty to understand guidelines provided in consultations, endorsed by the common loss of cognitive capacity typical of old age [36,48–50].

The analysis of the clinical results showed a greater frequency of patients with altered arterial pressure in the omeprazole group, with a statistically significant difference between the groups (p-value 0.016). Montenegro et al. conducted a randomized clinical trial and identified that PPI are capable of interfering with the activation and bioavailability of oral nitrite, thus promoting an increase in nitrite-dependent systemic blood pressure, since its activation depends on the acidic environment promoted by gastric acid [51] which could perhaps explain the increased blood pressure in this group. In addition, some studies have reported that magnesium plays a key role in blood pressure control and hypomagnesaemia, which may occur in the use of PPI. Magnesium plays a key role in blood pressure control, as it is responsible for regulating intracellular protein contraction and mediating calcium influx, which is responsible for the contraction of smooth muscle cells by activating protein kinase C. In smooth muscle cells present in vascular endothelium, magnesium plays a role as a calcium antagonist, inhibiting its transmembrane transport and thereby reducing calcium-dependent vasoconstrictor capacity [52,53]. Thus, hypomagnesemia may lead to increased calcium influx into vascular endothelial smooth muscle cells, causing vasoconstriction, further activation of cardiomyocytes, and a consequent increase in blood pressure [54,55]. In this sense, in addition to favoring the increase in blood pressure, which in itself is a factor that worsens the prognosis of patients with kidney disease, the use of PPI may further increase the risk of CKD onset and progression, since hypomagnesemia leads to increased secretion of atherogenic and inflammatory substances, resulting in endothelial dysfunction of the renal tissue [25]. Thus, effective control of serum magnesium levels in patients with CKD, being PPI users, is a fundamental factor for a better prognosis of these patients.

However, this association cannot be confirmed in the present study because the magnesium values of the patients were not analyzed. These results show the need for monitoring magnesium levels in PPI users since the literature points out that the use of these drugs for long periods can be related to the development of hypomagnesaemia, thus generating other complications in users, such as SAH, for example [16,25].

It is undeniable that PPI use can be a further risk factor for increasing blood pressure by decreasing oral nitrite bioavailability and reducing serum magnesium levels. However, especially in this group of patients, we should not lose sight of the fact that the determination of blood pressure in CKD patients is multifactorial because this is a complex physio-pathological context. To unequivocally establish this cause and effect relationship, more robust studies (such as randomized controlled trials) with control of all variables involved, are necessary. In addition, the population assisted at the nephrology ambulatory clinic had a median creatinine of 1.6 and an eGFR of 37.00, which reaffirmed a higher proportion of patients in stages 3a (31.1%) and 3b (35.7%) and the CKD stage may also be a direct factor determining blood pressure levels.

Concerning the comorbidities of the study population, DM II (39.7%) and SAH (89.4%) were the most prevalent. This result was already expected, since according to the last Brazilian Dialysis Census, these diseases remain as the main underlying causes of CKD [5]. Both are highly prevalent worldwide, with a high risk of mortality, and when uncontrolled, they are well-defined risk factors for the development of CKD and worsening of patient prognosis [4,17].

Regarding health-related behaviors, no significant statistical difference was found between users and non-users of omeprazole in relation to the variables evaluated. The majority of the patients followed-up at the nephrology ambulatory clinic in this study presented healthy life habits, since 82.6% were not alcohol consumers, 76.3% were non-smokers, and 66.8% were non-sedentary. In addition, 81.9% of them did not have dyslipidemia. In 2018, Kamal et al. (2018) reported that prolonged use of PPI is an isolated risk factor for the development of CVD [25]. Thus, the life habits of the patients in the present study represented a favorable finding and perhaps a positive influence on their prognosis. Although exposed to omeprazole, there were modifiable risk factors that could potentiate the worsening of renal function and consequently increase of the risk of CVD. However, these risk factors were controlled.

Another finding that seemed to favor the renal prognosis of the patients studied was the very low frequency of use of NSAIDs (3.5%). These drugs are known to be nephrotoxic and potentiated in the elderly [45,56,57]. This low frequency is probably justified by their known nephrotoxicity. In clinical practice, many prescribers use them with caution or avoid their prescription for CKD patients. Regarding nephroprotective drugs, ARBs were the most used (61.3%) followed by ACE inhibitors (21.6%). At least 61.3% of the CKD patients treated at the nephrology ambulatory clinic used at least one drug as a nephroprotection mechanism, which is a favorable profile of use for these patients.

Due to the characteristics of the study population (mostly elderly and with CKD), polypharmacy was common, as expected, both among users and non-users of omeprazole, with a median of seven and six medications, respectively. However, it was not possible to evaluate in this study the real benefit or harm of polypharmacy, since this requires another set of analysis of associations between the use of each drug, considering all the confounding variables, with the outcome of CKD evolution.

Returning to the discussion of renal impairment caused by regular use of PPI, the idea of recovering histamine receptor antagonists, predecessors to PPI used for the treatment of gastric diseases, has been raised as a more protective option to renal functions [8,9,58]. In recent meta-analysis, Wijarnpreecha et al. (2017) found a 1.3-fold increased risk for CKD onset and evolution to ESRD for PPI users. For the H2 antagonist users this risk was not found [59]. Xie et al. (2016) found in a cohort study that PPI users have a higher risk of CKD onset and evolution, as well as to progression to ESRD, when compared to users of H2 antagonists and controls [60]. Thus, these drugs emerge as a plausible alternative for the prevention and treatment of gastric diseases without the high risk of impaired renal function.

Finally, it is important to emphasize that despite the inestimable value of the findings of this study for clinical practice, it is necessary to recognize some limitations. Although this was a cohort study, a retrospective cohort design with use of secondary data (medical records and the health information system) favors biases such as selection and information typical of retrospective cohorts [61]. However, the selection bias was minimized by using the entire ambulatory population of CKD patients, thus preventing further omeprazole users who evolved from being selected. Regarding information bias, it is extremely important to consider possible errors in registration data of both sources. To reduce their occurrence, two data sources were used, which when compared, allowed the confirmation and complementation of missing data. In addition, double entry was made in the database in order to reduce the chances of registration errors of the collected data. Furthermore, the analyses were made with drug classes and with the total number of drugs in use, but the drugs were not analyzed individually. The evaluated drugs are commonly used in self-medication, and this prevents the evaluation of possible interference in these cases based on the use of secondary data. In the case of some variables, a large amount of missing data was detected at the time of data collection. This made it impossible to use such variables in the analyses of this study. Prior to the removal of these variables, an

analysis was performed to verify the possibility that they could be confounding factors, being confirmed as not influencing the final result. In this sense, a need to raise awareness among primary care professionals about the importance of properly completing the information required in health information systems was found. Despite these limitations, the quality of this study and the strength of the association found are not impaired. Furthermore, no statistically significant differences were found between the groups in most of the variables evaluated, reflecting high homogeneity between the groups evaluated, allowing a more reliable comparison between them.

The findings of this study are extremely important, since they contribute to the change of the approach in clinical practice. They also corroborate with results found in previous studies regarding the nephrotoxicity of PPI and their relation to CKD. The study alerts prescribers about the importance of rational use of PPI in patients without renal impairment, as well as the importance of avoiding them whenever possible in the therapy of CKD patients. Finally, the study contributes to the elaboration of updated guidelines for the adequate pharmacological treatment of elderly and/or CKD patients.

## Conclusion

There was a significant association between regular use of omeprazole and progression to worse stages of CKD in adult and elderly individuals followed-up at the nephrology ambulatory clinic at the Municipal Polyclinic of Divinópolis. Appropriate guidelines for health professionals about PPI nephrotoxicity need to be developed in order to preserve the renal function in elderly and/or CKD patients.

## Supporting information

**S1 Database.**
(XLSX)

## Acknowledgments

To Federal University of São João Del-Rei and Nephrology Ambulatory of Municipal Polyclinic of Divinopolis for promoting the development of this study.

## Author Contributions

**Conceptualization:** João Victor Marques Guedes, Jéssica Azevedo Aquino, Tássia Lima Bernardino Castro, Flávio Augusto de Morais, André Oliveira Baldoni, Vinícius Silva Belo, Alba Otoni.

**Data curation:** João Victor Marques Guedes, Jéssica Azevedo Aquino, Tássia Lima Bernardino Castro, Flávio Augusto de Morais, André Oliveira Baldoni, Vinícius Silva Belo, Alba Otoni.

**Formal analysis:** João Victor Marques Guedes, Jéssica Azevedo Aquino, Tássia Lima Bernardino Castro, Flávio Augusto de Morais, André Oliveira Baldoni, Vinícius Silva Belo, Alba Otoni.

**Investigation:** João Victor Marques Guedes, Jéssica Azevedo Aquino, Tássia Lima Bernardino Castro, Flávio Augusto de Morais, André Oliveira Baldoni, Vinícius Silva Belo, Alba Otoni.

**Methodology:** João Victor Marques Guedes, Jéssica Azevedo Aquino, Tássia Lima Bernardino Castro, Flávio Augusto de Morais, André Oliveira Baldoni, Vinícius Silva Belo, Alba Otoni.

**Project administration:** João Victor Marques Guedes, Jéssica Azevedo Aquino, Flávio Augusto de Morais, André Oliveira Baldoni, Vinícius Silva Belo, Alba Otoni.

**Resources:** Flávio Augusto de Morais, André Oliveira Baldoni, Vinícius Silva Belo, Alba Otoni.

**Supervision:** João Victor Marques Guedes, Tássia Lima Bernardino Castro, Flávio Augusto de Morais, André Oliveira Baldoni, Vinícius Silva Belo, Alba Otoni.

**Validation:** João Victor Marques Guedes, Jéssica Azevedo Aquino, Tássia Lima Bernardino Castro, Flávio Augusto de Morais, André Oliveira Baldoni, Vinícius Silva Belo, Alba Otoni.

**Visualization:** João Victor Marques Guedes, Jéssica Azevedo Aquino, Tássia Lima Bernardino Castro, Flávio Augusto de Morais, André Oliveira Baldoni, Vinícius Silva Belo, Alba Otoni.

**Writing – original draft:** João Victor Marques Guedes, Jéssica Azevedo Aquino, Tássia Lima Bernardino Castro, Flávio Augusto de Morais, André Oliveira Baldoni, Vinícius Silva Belo, Alba Otoni.

**Writing – review & editing:** João Victor Marques Guedes, Jéssica Azevedo Aquino, Tássia Lima Bernardino Castro, Flávio Augusto de Morais, André Oliveira Baldoni, Vinícius Silva Belo, Alba Otoni.

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
