## [Decision Letter · Decision Letter 0]

13 Sep 2019

PONE-D-19-23732

Omeprazole use and risk of chronic kidney disease evolution

PLOS ONE

Dear Mr. Guedes,

Thank you for submitting your manuscript to PLOS ONE. After careful consideration, we feel that it has merit but does not fully meet PLOS ONE’s publication criteria as it currently stands. Therefore, we invite you to submit a revised version of the manuscript that addresses the points raised during the review process.

The manuscript focuses on a topic of potential interest. However, the manuscript shows major shortcomings that should be addressed. To mention few of them, i) concern about the fact that it appears that there was a single evaluation as to whether or not the patients were on a PPI or not at the baseline visit; ii) need to perform analysis using PPI use as the time-varying variable; iii) need to specify the definition of PPI use in term of dose and duration; iv) concern about the use and definition of CKD stage evolution as outcome variable which is inadequate/incomplete; v) perform additional analysis on rapid progression; vi) clearly define the baseline creatinine; vii) need to further discuss and in more depth the association between PPI use and CKD; viii) unclear whether the investigators excluded kidney transplant patients; ix) need to follow the STROBE checklist; x) discuss the limitations of the findings of the study; xi) unclear whether the her blood pressure in the group with PPI than the group without PPI has been taken into account in the multivariate analysis.

We would appreciate receiving your revised manuscript by Oct 28 2019 11:59PM. To enhance the reproducibility of your results, we recommend that if applicable you deposit your laboratory protocols in protocols.io, where a protocol can be assigned its own identifier (DOI) such that it can be cited independently in the future. For instructions see: http://journals.plos.org/plosone/s/submission-guidelines#loc-laboratory-protocols

We look forward to receiving your revised manuscript.

Kind regards,

Giuseppe Remuzzi

Academic Editor

PLOS ONE

Journal Requirements:

1. We note that you have indicated that data from this study are available upon request. PLOS only allows data to be available upon request if there are legal or ethical restrictions on sharing data publicly. For information on unacceptable data access restrictions, please see http://journals.plos.org/plosone/s/data-availability#loc-unacceptable-data-access-restrictions.

Reviewers' comments:

Reviewer's Responses to Questions

**Comments to the Author**

1. Is the manuscript technically sound, and do the data support the conclusions?

Reviewer #1: Partly

Reviewer #2: Yes

2. Has the statistical analysis been performed appropriately and rigorously? 

Reviewer #1: No

Reviewer #2: I Don't Know

3. Have the authors made all data underlying the findings in their manuscript fully available?

Reviewer #1: Yes

Reviewer #2: Yes

4. Is the manuscript presented in an intelligible fashion and written in standard English?

Reviewer #1: No

Reviewer #2: Yes

5. Review Comments to the Author

Reviewer #1: It is an interesting and relevant article. I consider it a useful contribution in its field of nephrology; in which I agree with the authors that no prior study has been capable of identifying an association between the use of PPI by CKD patients with and the progression of this disease.

However, there are major modifications and clarifications that are strongly needed for this study. Avoidance of modifications to improve manuscript/study is discouraged.

1. It appears that there was a single evaluation as to whether or not the patients were on a PPI or not at the baseline visit. Is it assumed that all patients who were documented as being on a PPI at baseline were on PPIs permanently and that none of those in the non-PPI group were ever started on a PPI? PPI use can change over time. Can you perform analysis using PPI use as the time-varying variable? Lack of longitudinal data on PPI use - this appears to be based on documented PPI intake at a single time point. Thus, consider performing analysis using PPI use as the time-varying variable

2. Please specify the definition of PPI use in term of dose and duration

3. The use and definition of CKD stage evolution as Outcome variable is inadequate/incomplete. In addition to Decline in GFR category, the authors/investigators need to look into “A certain drop in eGFR is defined as a drop in GFR category accompanied by a 25% or greater drop in eGFR from baseline”. Thus, 25% or greater drop in eGFR from baseline should be taken into consideration

4. Additional analysis is needed on Rapid progression in which is defined as a sustained decline in eGFR of more than 5 mL/1.73m2/year, as KDIGO recommended.

5. In order to perform 1) and 2) for validity, baseline creatinine definition will need to be clear defined. Most recent vs. minimum baseline creatinine? When baseline creatinine is not available, SCrGFR-75 is suggested, when baseline outpatient SCr was not available or may need to exclude those patients from the study. Need clearer describe in the method.

6. Association between PPI use and CKD needs better discussed. In fact, recent meta-analysis showed 1.3-fold increased risks of CKD and ESRD in patients using PPIs, but not in patients using H2RAs. PMID: 28836158

7. RE “some studies have reported that magnesium plays a key role in blood pressure control and hypomagnesaemia, which may occur in the use of PPI, a possible cause of arterial hypertension resulting from increased calcium influx in smooth muscle cells of the vascular endothelium”; need better discussion; this statement is unclear. The use of PPIs has been shown to be associated with 1.43-fold increased risk of hypomagnesemia compared to those who did not use PPIs. PMID: 26108134. The underlying mechanism of the association of hypomagnesemia in patients with PPI use is likely explained by the disturbance of gastrointestinal (GI) handling of Mg since studies have shown that an increased renal Mg loss is not the only culprit in those patients with significant hypomagnesemia after PPI use.

8. Did the investigator exclude kidney transplant patients?

9. The STROBE checklist is needed to be followed. There are a number of consistent inaccuracies in the manuscript that I strongly believe need to be addressed before the paper can be reviewed again.

10. Please discuss in the limitations the types of biases the findings of this study may be influenced by.

11. Some revision of the English language is needed. There are some parts of the paper where it is quite difficult to make sense of some sentences. English edit will help to improve the quality of the manuscript.

“they identified that PPI are associated” is not correct in grammar.

RE “medicine use were collected from all patients diagnosed with CKD”; “medicine use” should be “medication use”

RE “the progressive decrease of renal function” “of” should be “in”

“In addition, considering the high consumption of PPI in the world and 78 in the Brazilian population [14] and their adverse effects on renal functions, perform research to 79 contribute to the reduction of the knowledge gap with respect to this association, and of the very use of 80 these drugs, is paramount.” Is not well written and difficult to follow.

Reviewer #2: Dear author,

Thanks for interesting article that brings new evidence of the renal toxicity of the proton pump inhibitor.

I would like to clarify a few points:

> major revision

The group with proton pump inhibitor has a higher blood pressure than the group without proton pump inhibitor. This difference alone can explain the progression of renal failure. it is not clear that this has been taken into account in the multivariate analysis.

> minor revision

in the table on page 10, for which data with ACEs do not appear?

> This is not the first time a study has focused on the progression of kidney failure. For example, XIE Y JASN 2016 has demonstrated a link between proton pump inhibitor and 30% decrease in GFR or doubling of creatinine level.

6. PLOS authors have the option to publish the peer review history of their article (what does this mean?). If published, this will include your full peer review and any attached files.

Reviewer #1: No

Reviewer #2: Yes: Desbuissons Geoffroy

---

## [Author Response · Author response to Decision Letter 0]

29 Oct 2019

October 27th, 2019.

Dear Editor Giuseppe Remuzzi,

Initially, we would like to thank you and the reviewers for their kind comments that helped us to improve our paper. We are resubmitting the revised manuscript for re-evaluation. We hope these revisions are satisfactory. If you require any further information please do not hesitate to contact us.

We inform that the manuscript has been modified according to the templates provided. Also, we would like to point out that the database was attached containing only the data necessary for the reproduction of the results obtained, in order to ensure the confidentiality of patients, as recommended by the Brazilian legislation on research ethics with human beings (RESOLUÇÃO Nº 510, DE 07 DE ABRIL DE 2016).

All statements made by the editor and the reviewers were considered in the revision of this manuscript, and answered as follows:

Reviewer #1: It is an interesting and relevant article. I consider it a useful contribution in its field of nephrology; in which I agree with the authors that no prior study has been capable of identifying an association between the use of PPI by CKD patients with and the progression of this disease.

However, there are major modifications and clarifications that are strongly needed for this study. Avoidance of modifications to improve manuscript/study is discouraged.

1. It appears that there was a single evaluation as to whether or not the patients were on a PPI or not at the baseline visit. Is it assumed that all patients who were documented as being on a PPI at baseline were on PPIs permanently and that none of those in the non-PPI group were ever started on a PPI? PPI use can change over time. Can you perform analysis using PPI use as the time-varying variable? Lack of longitudinal data on PPI use – this appears to be based on documented PPI intake at a single time point. Thus, consider performing analysis using PPI use as the time-varying variable.

Answer: The following sentence has been included in the text for further clarification:

“Initially, users of omeprazole were considered as those who at baseline (T0) had used this drug previously for at least three months. Thereafter all patients defined as users were evaluated for the other times (T1, T2, T3, T4) for the use of omeprazole, and only those who maintained its use during all evaluation times for at least two years were included. Follow-up data on drug use were available for all follow-up times. Only patients who continued to use the drug during all assessment times were considered exposed to omeprazole. In the two years of follow-up, there was no change in the prescription for any of the patients included, maintaining the daily dose of 20 mg/ day. Those who discontinued omeprazole during follow-up were not included in the study.”

2. Please specify the definition of PPI use in term of dose and duration

Answer: As presented above information regarding the omeprazole use, dose and duration are described in the exposure variable section and were adequate for better understanding:

“Initially, users of omeprazole were considered as those who at baseline (T0) had used this drug previously for at least three months. Thereafter all patients defined as users were evaluated for the other times (T1, T2, T3, T4) for the use of omeprazole, and only those who maintained its use during all evaluation times for at least two years were included. Follow-up data on drug use were available for all follow-up times. Only patients who continued to use the drug during all assessment times were considered exposed to omeprazole. In the two years of follow-up, there was no change in the prescription for any of the patients included, maintaining the daily dose of 20 mg/ day. Those who discontinued omeprazole during follow-up were not included in the study.”

3. The use and definition of CKD stage evolution as Outcome variable is inadequate/incomplete. In addition to Decline in GFR category, the authors/investigators need to look into “A certain drop in eGFR is defined as a drop in GFR category accompanied by a 25% or greater drop in eGFR from baseline”. Thus, 25% or greater drop in eGFR from baseline should be taken into consideration

Answer: As proposed by the reviewer, in addition to investigating the CKD evolution by changing to a worse stage of the disease, we also assessed a reduction in eGFR ≥ 25% for all follow-up times in users and non-users of omeprazole. Thus, the following excerpts were inserted in the manuscript:

“[...] In addition to this analysis, for each time, a 25% or more reduction in eGFR compared to the eGFR of the previous time was evaluated.” (Methods section)

[...] “Confirming a higher CKD evolution in omeprazole users, a decrease in eGFR ≥ 25% was found in omeprazole users more frequently than in non-users, and this difference was statistically significant (p = 0.002).” (Results)

4. Additional analysis is needed on Rapid progression in which is defined as a sustained decline in eGFR of more than 5 mL/1.73m2/year, as KDIGO recommended.

Answer: As suggested by the reviewer, CKD rapid progression analysis was performed, finding no statistically significant differences between users and non-users of omeprazole (p = 0.233). Thus, the following excerpts were inserted into the manuscript:

“Plus, a rapid progression analysis (5 mL/1.73m2/year) was performed as recommended by KDIGO [4].” (Methods: Outcome variable section)

“Regarding the rapid progression of CKD (5 mL/1.73m2/year), no statistically significant differences were found between users and non-users of omeprazole (p = 0.233).” (Results Section)

5. In order to perform 1) and 2) for validity, baseline creatinine definition will need to be clear defined. Most recent vs. minimum baseline creatinine? When baseline creatinine is not available, SCrGFR-75 is suggested, when baseline outpatient SCr was not available or may need to exclude those patients from the study. Need clearer describe in the method.

Answer: As thought by the reviewer, patients who did not have creatinine values at baseline or sequential times studied were excluded from the study. This information is described in Methods: Study design and population. Serum creatinine basal values as well as serum creatinine values from subsequent times were described in the medical records and were used to calculate eGFR, and thus to classify the CKD stage for each time. In this regard, in order to clarify this information, the following sentence has been inserted in the text:

“Patients without baseline creatinine values (T0) were excluded from the study because it would not be possible to determine the estimated glomerular filtration rate (eGFR) and thus the baseline CKD stage. Similarly, those patients who had not recorded serum creatinine values at subsequent times (T1, T2, T3 and T4) were also excluded.”

6. Association between PPI use and CKD needs better discussed. In fact, recent meta-analysis showed 1.3-fold increased risks of CKD and ESRD in patients using PPIs, but not in patients using H2Ras. PMID: 28836158

Answer: To improve the discussion about the association between the use of PPI and CKD, there was a discussion about the possible mechanisms associated with renal injury, comparing the findings regarding the use of PPIs with H2 antagonists, clarifying aspects concerning this association. Results that explain these findings were explained using the reference cited, as well as others for complementation. The following excerpts have been inserted:

“Investigating the possible mechanisms associated with renal damage caused by the use of PPI, Kamal et al. (2018) identified that the use of these drugs may be associated with the installation of a chronic process arising from a recurrent acute process, as well as the installation of the chronic process independent of the installation of AKI. In this sense, the authors propose some mechanisms that may be involved: the deposition of the drug and its metabolites in renal tissue, which may culminate in renal interstitial fibrosis, leading to chronicity of the lesion and onset of CKD; reduced nitric oxide synthesis, caused by inhibition of the proton pump of cell lysosomes, and thus production of highly reactive superoxide anion, which causes renal endothelial dysfunction; and hypomagnesemia, as low magnesium levels are able to increase secretion of atherogenic and inflammatory substances, producing endothelial dysfunction of the renal tissue [25].”

“In contrast to that presented for PPI, no results have yet been found that associate the use of H2 antagonists with the installation and/or progression of CKD. Some previous studies have suggested a possible association between the use of these drugs and the setting up of acute processes [26,27], but these results are still controversial, as current studies have not found an association between the use of these drugs and the installation of acute or chronic processes [8, 9, 28]. Possibly this difference found between the use of PPI and H2 antagonists may be related to their respective mechanisms of action, since they are of different drug classes and trigger different physiological effects.”

“Regarding the deposition of drugs and their metabolites in renal tissue, H2 antagonists have already been mentioned, and possibly by this mechanism could be associated with acute processes, however, it should be noted that at initially a large number of drugs could trigger an inflammatory response in renal tissue [27]. Regarding nitric oxide synthesis, there are no studies to date that found an association between the reduction of its synthesis and H2 antagonists, an expected result since these drugs do not act by inhibiting proton pumps, but by blocking histamine receptors. (H2). Regarding hypomagnesemia, a population study conducted by Markovits et al (2014) found that the use of H2 antagonists is associated with mild hypomagnesemia, only in use for long periods, and found no association of these drugs with moderate or severe hypomagnesemia. For PPI, a significant association was found with a higher association strength for mild, moderate and severe hypomagnesemia [29]. Thus, it can be concluded that probably the mechanisms leading to the onset and progression of CKD by the use of PPI are associated with the onset of hypomagnesemia, as already proposed by Kamal et al. (2018) [25]. In fact, a variety of studies have found a significant association between PPI use and hypomagnesemia [28-32].”

“Hypomagnesemia associated with the use of PPI may be partly explained by the renal loss of magnesium due to decreased resorption as well as decreased absorption, as PPI are capable of altering the pH of the gastrointestinal tract, thereby decreasing absorption mediated by TRPM6 and TRPM7 transporters [32].”

7. RE “some studies have reported that magnesium plays a key role in blood pressure control and hypomagnesaemia, which may occur in the use of PPI, a possible cause of arterial hypertension resulting from increased calcium influx in smooth muscle cells of the vascular endothelium”; need better discussion; this statement is unclear. The use of PPIs has been shown to be associated with 1.43-fold increased risk of hypomagnesemia compared to those who did not use PPIs. PMID: 26108134. The underlying mechanism of the association of hypomagnesemia in patients with PPI use is likely explained by the disturbance of gastrointestinal (GI) handling of Mg since studies have shown that an increased renal Mg loss is not the only culprit in those patients with significant hypomagnesemia after PPI use.”

Answer: In order to better elucidate a higher frequency of altered blood pressure found in PPIs users in the univariate analysis, we used studies to find a possible association between the use of these drugs, hypomagnesemia, blood pressure and CKD. For a better discussion of this, the following excerpt has been inserted into the manuscript:

“[…] Magnesium plays a key role in blood pressure control, as it is responsible for regulating intracellular protein contraction and mediating calcium influx, which is responsible for the contraction of smooth muscle cells by activating protein kinase C. In smooth muscle cells present in vascular endothelium, magnesium plays a role as a calcium antagonist, inhibiting its transmembrane transport and thereby reducing calcium-dependent vasoconstrictor capacity [52,53]. Thus, hypomagnesemia may lead to increased calcium influx into vascular endothelial smooth muscle cells, causing vasoconstriction, further activation of cardiomyocytes, and a consequent increase in blood pressure [54,55]. In this sense, in addition to favoring the increase in blood pressure, which in itself is a factor that worsens the prognosis of patients with kidney disease, the use of PPI may further increase the risk of CKD onset and progression, since hypomagnesemia leads to increased secretion of atherogenic and inflammatory substances, resulting in endothelial dysfunction of the renal tissue [25]. Thus, effective control of serum magnesium levels in patients with CKD, being PPI users, is a fundamental factor for a better prognosis of these patients.”

“[…] In fact, a variety of studies have found a significant association between PPI use and hypomagnesemia [28-32].”

“Hypomagnesemia associated with the use of PPI may be partly explained by the renal loss of magnesium due to decreased resorption as well as decreased absorption, as PPI are capable of altering the pH of the gastrointestinal tract, thereby decreasing absorption mediated by TRPM6 and TRPM7 transporters [32].”

8. Did the investigator exclude kidney transplant patients?

Answer: All patients with kidney transplantation and / or on RRT were not included in this study. To make this information clearer, we added this information to the exclusion criteria:

“[...] Patients who underwent previous kidney transplantation and/or patients on renal replacement therapy, and patients who had missing data related to some of the variables used during the two-year follow-up were not included in this study.” (Methods section)

9. The STROBE checklist is needed to be followed. There are a number of consistent inaccuracies in the manuscript that I strongly believe need to be addressed before the paper can be reviewed again.

Answer: As proposed by the reviewer the STROBE checklist was revised after all modifications. Below is the properly completed STROBE checklist, demonstrating that all steps to properly write this manuscript have been checked:

10. Please discuss in the limitations the types of biases the findings of this study may be influenced by.

As described in the discussion although limitations have been identified, for each of them alternatives were worked out to minimize possible interference in the final result: 

• Selection bias was minimized using the entire outpatient population (users and non-users) of CKD patients, thus preventing the selection of only omeprazole users who evolved to CKD.

• Regarding the information bias, two sources (medical records and SIS data) were used to validate all collected data. Thus, the results represent, in fact, the reality of patients with the lowest possible risk of information errors. 

• Regarding drug classes: we worked with the total number of drugs in use per patient. This way of working was determined to minimize the impacts of those unregistered medications. 

• Some variables with a large amount of missing data detected at the time of data collection, so that there were no results induced by the absence of data, were withdrawn from the study. Assessed the impact of each variable of being a confounding variable before withdrawal, thus being confirmed as not influencing the final outcome.

11. Some revision of the English language is needed. There are some parts of the paper where it is quite difficult to make sense of some sentences. English edit will help to improve the quality of the manuscript.

“they identified that PPI are associated” is not correct in grammar.

RE “medicine use were collected from all patients diagnosed with CKD”; “medicine use” should be “medication use”

RE “the progressive decrease of renal function” “of” should be “in”

“In addition, considering the high consumption of PPI in the world and 78 in the Brazilian population [14] and their adverse effects on renal functions, perform research to contribute to the reduction of the knowledge gap with respect to this association, and of the very use of these drugs, is paramount.” Is not well written and difficult to follow. 

Answer: For a better clarification of the text, the sentences have been corrected as follows:

“However, recent studies found that loss of renal function is not necessarily caused by sequential acute lesions, as the use of PPI has been associated with CKD regardless of the occurrence of previous acute episodes [8,10].

“Information about clinical and sociodemographic data, health behaviors, and medication use was collected from all patients diagnosed with CKD through consultation of medical charts and the Brazilian health information system (SIS).”

“Chronic kidney disease (CKD), characterized by progressive deterioration of biochemical and physiological functions of the body systems, can be defined as a syndrome caused by the progressive decrease in renal function [1-4].”

“In addition, considering the high consumption of PPI in Brazil [14] and in the world population, as well their adverse effects, it is necessary to conduct researches to better elucidate this association.”

In order to guarantee the application of English grammar standards, this manuscript has been reviewed by a qualified professional. Thus, minor corrections were also made throughout the text, highlighted in yellow. Attached is proof of revision:

Reviewer #2: 

Dear author,

Thanks for interesting article that brings new evidence of the renal toxicity of the proton pump inhibitor.

I would like to clarify a few points:

major revision:

The group with proton pump inhibitor has a higher blood pressure than the group without proton pump inhibitor. This difference alone can explain the progression of renal failure. It is not clear that this has been taken into account in the multivariate analysis.

Answer: In fact, blood pressure could be a factor that could explain CKD progression, since hypertension is one of the main underlying diseases, and also one of the factors that worsen prognosis when not controlled. Although a statistically significant difference was found when comparing the users and non-users of omeprazole groups by Pearson’s chi-square test, when performing univariate COX analysis to identify the variables eligible for the multivariate adjustment model, this variable did not presented statistical significance (p> 0.20), indicating that this is not a confounding variable in this study. Therefore, this variable was not included in the multivariate analysis. It is noteworthy that all variables were tested in the univariate model, and only the variables Omeprazole use (exposure) and ARBs use were statistically significant for inclusion in the multivariate model. To make this information better understood, the following text was added to the manuscript:

“The use of the other drugs evaluated (NSAIDs, ACE inhibitors, ARBs) and blood pressure presented p-value> 0.20 in the univariate Cox analysis, showing that these are not confounding variables and, therefore, not included as adjustment variables in the multivariate analysis.”

>minor revision:

in the table on page 10, for which data with ACEs do not appear?

R: As described above, this variable was not statistically significant in the univariate COX analysis (p> 0.20), and therefore was not included in the multivariate analysis.

> This is not the first time a study has focused on the progression of kidney failure. For example, XIE Y JASN 2016 has demonstrated a link between proton pump inhibitor and 30% decrease in GFR or doubling of creatinine level.

R: In fact, the article discusses the evolution of CKD in patients using PPIs, so it was rescued by the authors and used to make the discussion of the use of PPIs more robust and the evolution of kidney disease. In addition, this information has been corrected in the Introduction section. Thus, the following excerpt was inserted in the discussion section:

“In recent meta-analysis, Wijarnpreecha et al. (2017) found a 1.3-fold increased risk for CKD onset and evolution to ESRD for PPI users. For the H2 antagonist users this risk was not found [59]. Xie et al. (2016) found in a cohort study that PPI users have a higher risk of CKD onset and evolution, as well as to progression to ESRD, when compared to users of H2 antagonists and controls [60].”

With kind regards,

The authors.

---

## [Decision Letter · Decision Letter 1]

5 Feb 2020

Omeprazole use and risk of chronic kidney disease evolution

PONE-D-19-23732R1

Dear Dr. Guedes,

We are pleased to inform you that your manuscript has been judged scientifically suitable for publication and will be formally accepted for publication once it complies with all outstanding technical requirements.

**The revised manuscript is definitely improved. The authors have adequately addressed the reviewers’ comments.**

With kind regards,

Giuseppe Remuzzi

Academic Editor

PLOS ONE

Additional Editor Comments (optional):

Reviewers' comments:

Reviewer's Responses to Questions

**Comments to the Author**

1. If the authors have adequately addressed your comments raised in a previous round of review and you feel that this manuscript is now acceptable for publication, you may indicate that here to bypass the “Comments to the Author” section, enter your conflict of interest statement in the “Confidential to Editor” section, and submit your "Accept" recommendation.

Reviewer #1: All comments have been addressed

Reviewer #3: All comments have been addressed

2. Is the manuscript technically sound, and do the data support the conclusions?

Reviewer #1: Yes

Reviewer #3: Yes

3. Has the statistical analysis been performed appropriately and rigorously? 

Reviewer #1: Yes

Reviewer #3: Yes

4. Have the authors made all data underlying the findings in their manuscript fully available?

Reviewer #1: Yes

Reviewer #3: No

5. Is the manuscript presented in an intelligible fashion and written in standard English?

Reviewer #1: Yes

Reviewer #3: Yes

6. Review Comments to the Author

Reviewer #1: I have no competing interests. It appears that all comments have been appropriately responded to. I have no further comments and recommend publication.

Reviewer #3: Interesting, and confirms prior findings. I read the paper and the responses from the first round of reviews. The authors addressed the comments. I have no further

7. PLOS authors have the option to publish the peer review history of their article (what does this mean?). If published, this will include your full peer review and any attached files.

Reviewer #1: No

Reviewer #3: No

---

## [Editor Report · Acceptance letter]

7 Feb 2020

PONE-D-19-23732R1 

Omeprazole use and risk of chronic kidney disease evolution 

Dear Dr. Guedes:

I am pleased to inform you that your manuscript has been deemed suitable for publication in PLOS ONE. Congratulations! Your manuscript is now with our production department. 

With kind regards,

on behalf of

Prof. Giuseppe Remuzzi 

Academic Editor

PLOS ONE